# Knowledge transfer-driven estimation of knee moments and ground reaction forces from smartphone videos via temporal-spatial modeling of augmented joint kinematics

Md Sanzid Bin Hossain[1,2], Hwan Choi[3]*, Zhishan Guo[4], Sunyong Yoo[5], Min-Keun Song[6], Hyunjun Shin[7], Dexter Hadley[1]

1 Department of Clinical Science, University of Central Florida, College of Medicine, Orlando, Florida, United States of America, 2 Center for Data Science, Nell Hodgson Woodruff School of Nursing, Emory University, Atlanta, Georgia, United States of America, 3 Department of Mechanical and Aerospace Engineering, University of Central Florida, Orlando, Florida, United States of America, 4 Department of Computer Science, North Carolina State University, Raleigh, North Carolina, United States of America, 5 Department of Intelligent Electronics and Computer Engineering, Chonnam National University, Gwangju, Republic of Korea, 6 Department of Physical and Rehabilitation Medicine, Chonnam National University Medical School, Gwangju, Republic of Korea, 7 Korea Orthopedics Rehabilitation Engineering Center, Incheon, Republic of Korea

* hwan.choi@ucf.edu

**Data availability statement:** The public dataset used in this study is available at the project IMU

## Abstract

The knee adduction and flexion moment provides critical information about knee joint health, while 3D ground reaction forces (GRFs) help identify force and energy characteristics for maneuvering the entire human body. Existing methods of acquiring joint moments and GRFs require expensive equipment, time-consuming pre-processing, and limited accessibility. This study proposes to tackle these limitations by utilizing only smartphone videos to estimate joint moments and 3D GRFs accurately. We also propose the augmentation of joint kinematics by generating additional modalities of 2D joint center velocity and acceleration from 2D joint center position acquired from the videos. This augmented joint kinematics helps to apply a multi-modal fusion module to learn the importance of inter-modal interactions. Additionally, we utilize recurrent neural networks and graph convolutional networks to perform temporal-spatial modeling of joint center dynamics for enhanced accuracy. To overcome another challenge of video-based estimation, particularly the lack of inertial information related to body segments, we propose multi-modal knowledge transfer to train the video-only student model from a teacher model that integrates both video and inertial measurement unit (IMU) data. The student model significantly reduces the normalized root mean square error (NRMSE) from 5.71 to 4.68 and increases the Pearson correlation coefficient (PCC) from 0.929 to 0.951. These results demonstrate that knowledge transfer, augmentation of joint kinematics for multi-modal fusion, and temporal-spatial modeling significantly enhance smartphone

and Smartphone Camera Fusion for Estimation of Kinetic Outcomes, hosted on the SimTK repository (https://simtk.org/projects/imukinetics). A detailed implementation of the models, including state-of-the-art approaches, is available at the GitHub repository Smartphone-Video-based-KAM-KFM-3D-GRFs (https://github.com/Md-Sanzid-Bin-Hossain/Smartphone-based-Kinetics-Estimation).

**Funding:** This work was supported by the National Science Foundation, United States, under Grant FRR-2246671, 2246672. The funders had no role in study design, data collection and analysis, decision to publish, or preparation of the manuscript.

**Competing interests:** Authors declare no competing interests.

video-based estimation, offering a potential cost-effective alternative to traditional motion capture for clinical assessments, rehabilitation, and sports applications.

## Introduction

Human walking kinetics, including joint moments and 3D ground reaction forces (GRFs), provide crucial insights into locomotor function [1,2]. During dynamic tasks, excessive lateral joint moments, such as abduction and adduction, can increase the risk of injuries [3], including ligament tears and cartilage damage [4], particularly when combined with high-impact forces or improper landing mechanics [5]. Since these joint moments contribute to overall joint loading [6], which is influenced by external forces acting on the body [7], GRFs also play a crucial role in characterizing force production, impulse generation, and energy transfer during movement [8]. Therefore, joint moments and GRFs provide comprehensive information to help understand joint functions and injury mechanisms. Specifically, the knee joint is vulnerable to injury due to its complex ligament structure [9] and its multiple degrees of freedom [10], which include flexion-extension as the primary motion along with knee adduction moment (KAM) and knee flexion moment (KFM). For example, KAM and KFM were used to evaluate the risk of knee injuries for elders [11] and athletes [12], while GRFs have been extensively utilized to analyze abnormal gait patterns of individuals with neurological disorders [13] and lower limb amputees [14]. Given the clinical and biomechanical significance, developing cost-effective and accessible alternatives to the lab-based methods for estimating these kinetic parameters could significantly advance movement analysis, injury prevention, and rehabilitation.

Despite their importance, conventional methods for calculating KAM, KFM, and GRFs are typically restricted to costly laboratory setups, and this requires time-consuming postprocessing [15]. To overcome the limitations of conventional approaches, data-driven techniques with wearable sensors, specifically with inertial measurement units (IMUs), have gained popularity in the estimation of kinetic parameters [16–22]. Integration of 2-D joint center position data, extracted from the videos of smartphone cameras, with IMUs can enhance the accuracy of KAM and KFM estimation, increasing the accuracy solely through IMUs [23]. While IMUs might offer a cost-effective and reliable way of acquiring kinetic data, the need for multiple sensors [24,25] and their setup can increase the overall cost, effort, and time required. On the other hand, the smartphone has a high level of accessibility and can be effortlessly utilized by configuring the camera without requiring any prior subject preparation. Thus, cameras in smartphones have the potential to be seamlessly incorporated into home or clinical settings, hence facilitating prompt evaluation. However, smartphone-based estimation alone may not provide accurate outcomes compared to using IMUs [23]. While smartphone videos can provide 2D joint locations through a joint center detection algorithm such as OpenPose [26], they still cannot provide the necessary inertial data of each body segment for accurate kinetic estimations. In contrast, IMUs can measure the segment-level acceleration and angular velocities, which can then be used to predict joint moments. Therefore, the main technical challenge addressed in this study is how to utilize only the joint center position data to make the prediction outcome more accurate. To address this challenge, this study develops a novel deep learning model incorporating augmented joint kinematics, temporal-spatial modeling, and multi-modal knowledge transfer.

To overcome the reduced accuracy of smartphone-only approaches, we augment the extracted joint center position data by deriving additional dynamic features of joint center

velocities and accelerations to capture better motion characteristics. This enables us to convert the uni-modal problem into a multi-modal one by utilizing velocities and accelerations in addition to the corresponding joint center position data. Thus, the system can capture additional dynamic information that would be missed with positional data alone. To effectively utilize the complementary strengths of different input modalities, we customize and update a multi-modal fusion technique combining multiple fusion modules (MFM), based on our previous multi-modal fusion module to enhance accuracy outcomes [27]. Using multiple fusion modules enables the model to capture different aspects of the multi-modal data, which helps to capture complex and diverse interactions between modalities more effectively than using a single fusion module. Hence, our approach enables accurate estimations of the kinetic parameters, making it more robust and accurate than previous work [23] that used a single fusion module, such as low-rank multi-modal fusion (LMFN) or tensor fusion network (TFN).

Beyond multi-modal fusion of augmented joint kinematics data, understanding the spatial structure of joints could help the deep learning model to understand the input better. As each node of the hip, knee, and ankle joint centers is connected by lines, the data can be represented with a graph-like structure. This structure helps to model the spatial relationships between joints using a graph convolutional network (GCN) [28,29]. We also integrate an attention mechanism to identify and emphasize the most important joints. This attention-based GCN helps capture spatial relationships, while a bi-directional long-short-term memory (Bi-LSTM [30])-based encoder captures the temporal aspect of the input. Thereafter, we combine both encoders through a gating mechanism to control the flow of features, which ultimately leads to a more accurate outcome.

While the discussed methods improve the smartphone-only estimation, their accuracy can be further enhanced using knowledge transfer from additional modalities. Thus, we introduce a multi-modal knowledge transfer technique using the teacher and student models. By leveraging knowledge from the multi-modal input of IMUs and the smartphone-based deep learning model (teacher) and transferring it to only the smartphone-video-based student model through the knowledge transfer, the performance of smartphone-based estimation can be significantly enhanced. This approach enables smartphone-based methods to achieve higher accuracy, thereby boosting the potential of video-based approaches for capturing detailed motion dynamics effectively. To implement this, we develop a two-step knowledge transfer technique to train the student model more efficiently, surpassing the performance and training complexity of parameter optimization of the conventional knowledge distillation techniques. The overall process of our proposed method is presented in Fig 1.

**Novelty and contributions.** Bringing all the components together, our approach leverages augmented dynamics, temporal-spatial modeling, and multi-modal knowledge transfer to enable accurate, video-only kinetic estimation. More specifically, our proposed student model has three unique features: (1) it integrates joint center velocity and acceleration in addition to the joint center position and handles multi-modal fusion; (2) an attention-based GCN is employed with Bi-LSTM encoder with a gating mechanism to effectively model the spatial-temporal aspect of the data; and (3) the student model is enriched with the knowledge from a teacher model with comprehensive input modalities through multi-modal knowledge transfer. We provide comprehensive ablation studies to show the effectiveness of each proposed component. To the best of our knowledge, this is the first study that uses temporal-spatial modeling of the augmented joint kinematics and knowledge transfer to promote accurate smartphone-based kinetic parameter estimation.

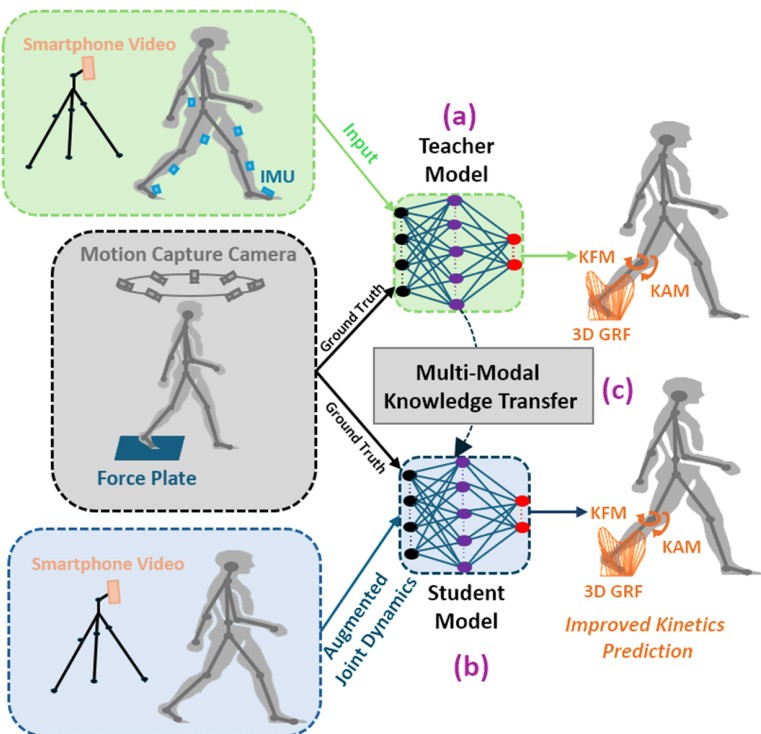

**Fig 1. Overview of the proposed multi-modal knowledge transfer approach.** (a) A teacher model, trained with input from inertial measurement units (IMUs) and smartphone videos, estimates knee adduction moment (KAM), knee flexion moment (KFM), and 3D ground reaction forces (GRFs); (b) a student model, trained with augmented joint kinematics derived from smartphone videos, aims to replicate these kinetic outputs; (c) the multi-modal knowledge transfer module enables the student model to benefit from the knowledge learned by the teacher model, resulting in improved kinetic outcomes. The teacher model workflow is shown in green, while the student model workflow is represented in blue.

## Methodology

### Problem statement

This paper estimates KAM, KFM, and 3D GRFs using videos from two smartphone cameras via a novel **student model**. If 2-D joint center position data derived from the smartphone videos are $\mathbb{V}^{jcp} \in \mathbb{R}^{N \times \Delta T \times (D_{video}^{jcp} \cdot M^{jcp})}$, we can derive 2-D joint center velocity $\mathbb{V}^{jcv} \in \mathbb{R}^{N \times \Delta T \times (D_{video}^{jcv} \cdot M^{jcv})}$ and acceleration $\mathbb{V}^{jca} \in \mathbb{R}^{N \times \Delta T \times (D_{video}^{jca} \cdot M^{jca})}$ by performing differentiation. Then, the kinetics estimation from the **student model** can be realized using Eq 1.

$$\mathbb{K}_{estimation}^{student} = f_{student}\left(\mathbb{V}^{jcv}, \mathbb{V}^{jca}, \mathbb{V}^{jcp}\right) \tag{1}$$

Here, $\mathbb{K}_{estimation}^{student} \in \mathbb{R}^{N \times \Delta T \times D_K}$, $\mathbb{K}_{estimation}^{student}$ has $N$ sequences, each with $\Delta T$ frames, and each frame has $D_K$ kinetic components.

Additionally, a **teacher model** is employed, where eight IMUs and videos from two smartphone cameras are used for estimation of KAM, KFM, and 3D GRFs, which later helps to transfer knowledge to the **student model** (Fig 1). If we have accelerometer and gyroscope data of $\mathbb{I}^{acc} \in \mathbb{R}^{N \times \Delta T \times (D_{imu}^{acc} \cdot M^{acc})}$ and $\mathbb{I}^{gyr} \in \mathbb{R}^{N \times \Delta T \times (D_{imu}^{gyr} \cdot M^{gyr})}$ and 2-D joint center position data $\mathbb{V}_{\Delta T}^{jcp} \in \mathbb{R}^{N \times \Delta T \times (D_{video}^{jcp} \cdot M^{jcp})}$ for a specific window length of $\Delta T$, then estimation from the

**teacher model** for the same window length of $\Delta T$, $\mathbb{K}^{teacher}_{estimation} \in \mathbb{R}^{N \times \Delta T \times D_K}$ can be realized by Eq 2.

$$\mathbb{K}^{\text{teacher}}_{\text{estimation}} = f_{\text{teacher}}\left(\mathbb{I}^{\text{acc}}, \mathbb{I}^{\text{gyr}}, \mathbb{V}^{\text{jcp}}\right) \tag{2}$$

Here, $D^{acc}_{imu}$ and $D^{gyr}_{imu}$ represent three-axis acceleration and angular velocity data, $D^{jcp}_{video} = D^{jcv}_{video} = D^{jca}_{video}$ represent the total number of joint center, $M^{acc}$ and $M^{gyr}$ are the number of IMUs, and $M^{jcp} = M^{jcv} = M^{jca}$ is the product of the total number of smartphone cameras that are employed to collect video and the dimension of the joint center position. $N$ represents the total number of samples in the dataset, where $N = B \times N_B$, B is the batch size and $N_B$ is the total number of batches during the model training. Here, $\Delta T = 50$, $D^{acc}_{imu} = D^{gyr}_{imu} = 3$, $D^{jcp}_{video} = D^{jcv}_{video} = D^{jca}_{video} = 11$, $M^{acc} = M^{gyr} = 8$, $M^{jcp} = M^{jca} = M^{jcv} = 4$.

## Dataset description

This study used a publicly available dataset of seventeen male subjects (age: 23.2 ± 1.1 years; height: 1.76 ± 0.06 m; mass: 67.3 ± 8.3 kg), approved by the Institutional Review Board of Shanghai Jiao Tong University under Application No. E2021101I and informed consent was obtained as documented in the original publication [23]. The dataset was accessed in January 2023 from its public GitHub repository. All data were de-identified prior to release, and the authors had no access to identifiable information at any point. Different testing conditions were tested, including different walking speeds, foot progression angles, step widths, and trunk sway angles. To establish a baseline for quantifying the foot progression angle and step width, subjects initially completed a 2-minute trial of normal walking at a self-selected speed (1.16 ± 0.04 m/s). Nine trials were conducted using combinations of three different foot progression angles (baseline–15°, baseline, and baseline+15°) and three different speeds (self-selected –0.2 m/s, self-selected, and self-selected +0.2 m/s). Similarly, trials involving three different step widths (baseline –0.054 m, baseline, and baseline +0.070 m) and trunk sway angles (4°, 8°, and 12°) were conducted, each at three different speeds.

**Dataset pre-processing:** An optical motion capture system (Vicon, Oxford Metrics Group, Oxford, UK) was used to collect the marker trajectories with a sampling rate of 100 Hz, while ground reaction force data were collected (Bertec Corp., Worthington, OH, USA) at 1000 Hz. Marker and ground reaction forces data are used to calculate KFM and KAM. Additionally, data from eight IMUs (SageMotion, Kalispell, MT, USA) were collected at a sampling rate of 100 Hz. Two smartphone cameras (iPhone 11, Apple Inc., Cupertino, CA, USA) were used to collect the video data during the data collection with a sampling rate of 120 Hz. Openpose (version 1.7.0, Body_25) [26] was used to estimate 2-D joint positions from each frame of the video for shoulder, hip, knee, and ankle joints. The origin was defined as the point midway between the left and right hip joints, allowing other 2-D joint positions to be independent of the subject's location in the global frame. The 2-D joint position was normalized with the corresponding subject's height to avoid the influence of subject size. In the pre-processed dataset provided in [23], GRFs and video data are resampled to synchronize with IMU, KFM, and KAM. KAM and KFM are provided in dimensionless form, normalized by body weight and body height, and expressed as percentages in the original dataset [23]. We normalized GRFs by subject body weight. These normalization strategies ensure that inter-subject variability due to body size or mass is reduced, enabling the deep learning model to learn subject-independent mappings between 2-D joint center input and kinetics output. Importantly, because the ground truth data are already normalized, the model does not require explicit knowledge of the subject's absolute body weight or height at inference. If subject-specific values in physical units are needed, they can be recovered post-prediction by rescaling the

normalized outputs with the individual's body weight and height. This approach balances generalizability across subjects while still allowing biomechanically meaningful scaling when subject-specific information is available. To ensure synchronization, zeros were added to the end of shorter steps in the dataset to match the length of the longest steps, which had a duration of 152 data points. Subsequently, we remove the extra zeros at the end of the shorter steps, leaving the remaining dataset to serve as the basis for constructing our dataset.

A more detailed description of the experimental setup and dataset processing can be found in [23].

## Proposed approach

The workflow for the algorithm development is divided into three stages. In the **first stage**, we utilize 2-D joint center position data acquired from smartphone videos to generate joint center velocity and acceleration, and use all these modalities together to build the multi-modal student model. In the **second stage**, we build a teacher model with multi-modal input such as IMUs and 2-D joint center position data. In the **third stage**, we leverage the proposed multi-modal knowledge transfer approach to enhance the student model's performance by incorporating knowledge from the teacher model.

**Multi-modal knowledge transfer.** We implement the multi-modal knowledge transfer framework in two steps. For the **first step**, we align the concatenated features of the student and teacher encoders. Specifically, we train the student encoder while minimizing root mean square error (RMSE) between the linearly projected encoded features ($X_{teacher}^{concat,lin}$) of the already trained teacher model and the linearly projected student model's encoded features $X_{student}^{concat,bi-lstm,gcn}$ (Fig 2). We apply a linear transformation (Eqs 3 and 4) to the concatenated features of Eqs 25 and 31 and use RMSE between these two as the feature alignment (FA) loss function to train the student encoder. In this step, we only utilize the encoder portion of the student model to build the pre-trained student encoder. In the **second step**, we add the multi-fusion module (MFM) and a fully connected layer with the pre-trained encoders (frozen weights) and train the full model (Fig 2).

$$X_{student}^{concat,lin} = FC(X_{student}^{concat,bi-lstm,gcn}) \tag{3}$$

$$X_{teacher}^{concat,lin} = FC(X_{teacher}^{concat}) \tag{4}$$

**Student model (augmented joint Kinematics+MFM+Gate+BiLSTM+Attn-GCN).** The student model is built with multiple components, including three encoders with a Bi-LSTM layer, an attention-based GCN layer, and a gated network to control the flow of information between these two. The MFM combines features from different modalities more effectively for performance improvement. We derive 2-D joint center velocity and acceleration from 2-D joint center position data to create additional modalities, which later help to improve the performance. Each component is described in the following subsections.

**a) Joint center position, velocity, and acceleration:** We leverage the 2D joint center position data and differentiate it to derive joint center velocity and acceleration, thereby generating additional modalities. The finite difference method is utilized to calculate velocity and acceleration, with the initial velocity set to zero and the first two and last points of the acceleration also set to zero to handle edge cases. Then, we can treat joint center position, velocity, and acceleration as separate modalities. Having more modalities can provide different aspects of the features and provide a more comprehensive representation. In this problem set, the joint position provides spatial information about the subject's joint center, and joint center velocity

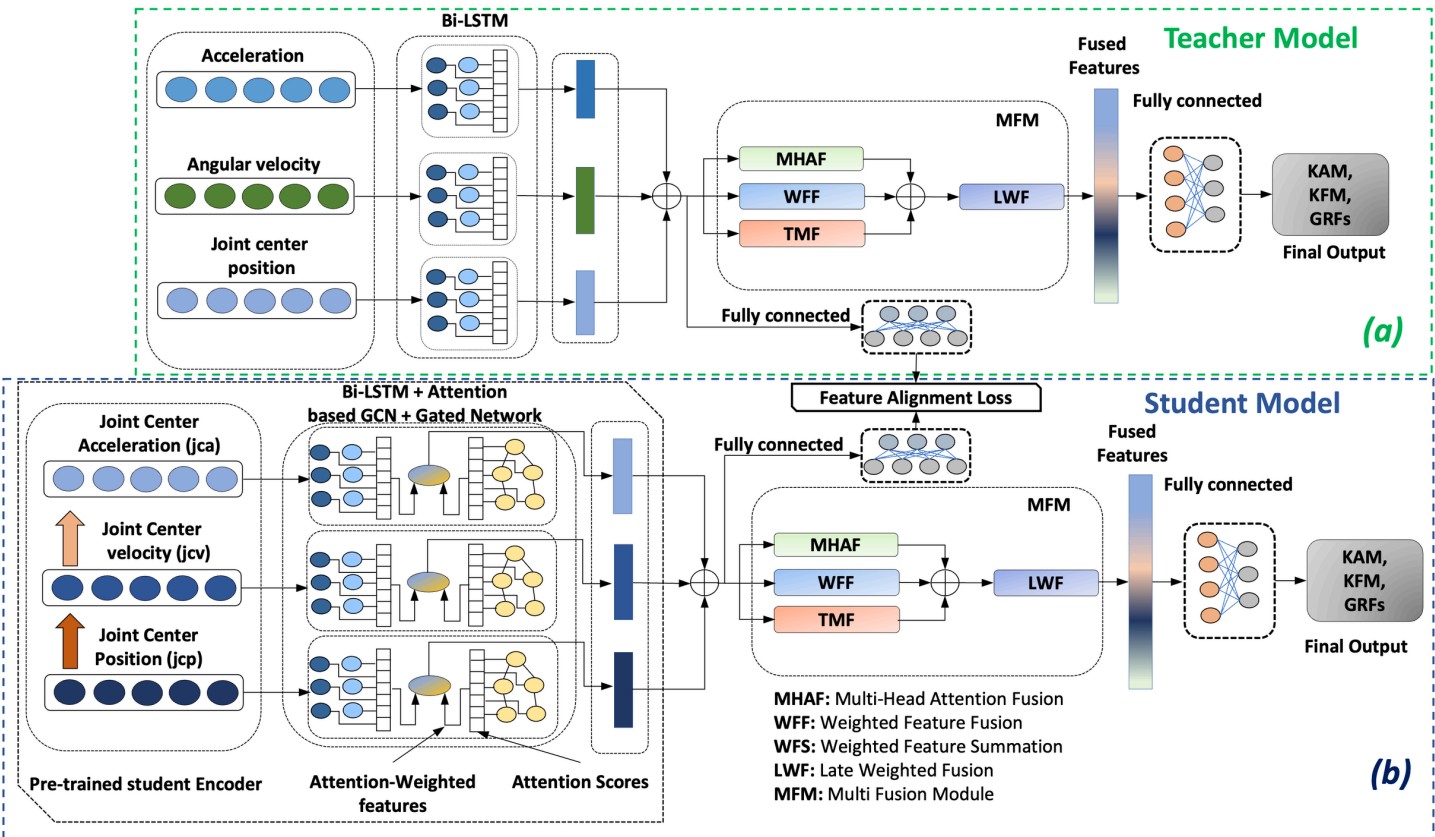

**Fig 2. Architecture of the proposed multi-modal knowledge transfer framework.** (a) The teacher model (green box) is trained using acceleration, angular velocity, and joint center position data through a Bi-LSTM encoder, followed by multi-fusion module (MFM) and fully connected layers to estimate KAM, KFM, and GRFs. (b) The encoder of the student model (blue box) is pre-trained using feature alignment (FA) loss by leveraging the trained teacher encoder while using augmented joint kinematics as input. The student encoder is then coupled with MFM and fine-tuned to obtain the final model.

and acceleration provide additional dynamic aspects of the subject's movement. As we estimate joint moments and GRFs, which are closely related to the subject's dynamics, these additional modalities, while treating them separately, may help to enhance the performance of the model.

**b) Encoder (Bi-LSTM+Attention based GCN+Gated Network:)** The encoder of the student model consists of a Bi-LSTM layer, an attention-based GCN layer, and a gated network to control information between these two layers.

**Bi-LSTM:** Two Bi-LSTM layers are stacked together to encode features from each modality. We apply a dropout layer after each Bi-LSTM layer to avoid overfitting. A batch normalization layer is applied after each modality to perform an operation similar to standard normalization [31] before being put as input to the Bi-LSTM layer. If the batch normalized features from joint center velocity, acceleration and position are $V^{jcv} \in \mathbb{R}^{B \times \Delta T \times (D^{jcv}_{video} \cdot M^{jcv})}$, $V^{jca} \in \mathbb{R}^{B \times \Delta T \times (D^{jca}_{video} \cdot M^{jcv})}$, and $V^{jcp} \in \mathbb{R}^{B \times \Delta T \times (D^{jcp}_{video} \cdot M^{jcp})}$, respectively. The output from the Bi-LSTM encoder can be found using Eqs 5, 6, and 7.

$$X^{jcv}_{bi-lstm} = Bi-LSTM(V^{jcv}) \tag{5}$$

$$X^{jca}_{bi-lstm} = Bi-LSTM(V^{jca}) \tag{6}$$

$$X^{jcp}_{bi-lstm} = Bi-LSTM(V^{jcp}) \tag{7}$$

Here $X_{bi-lstm}^{jcv}, X_{bi-lstm}^{jca}, X_{bi-lstm}^{jcp} \in \mathbb{R}^{B \times \Delta T \times C}$, where $C$ is the dimension of output features. Then, we concatenate features from three modalities to fuse all the multi-modal features together (Eq 8).

$$X_{student}^{concat,bi-lstm} = \left[ X_{bi-lstm}^{jcv}, X_{bi-lstm}^{jca}, X_{bi-lstm}^{jcp} \right] \tag{8}$$

**Attention-based GCN:** Human joint centers can be represented as a graph-like structure. We utilize this structure to derive additional features using a GCN, which can help to model the spatial relationship between joint center data. These additional features can improve the results of kinetics prediction. Moreover, we apply an attention mechanism to assign more weight to the node that helps to estimate the kinetic parameters. Specifically, as we are estimating the KFM, KAM, and GRFs of the right leg, the model should put more weight on the node that builds the right leg of the skeleton.

As the input size of GCN and Bi-LSTM is different, we need to reshape the batch-normalized input features so that they fit as the input of GCN. If the batch normalized features from joint center velocity, acceleration, and position are $V_{gcn}^{jcv} \in \mathbb{R}^{(B \cdot \Delta T) \times D_{video}^{jcv} \times M^{jcv}}$, $V_{gcn}^{jca} \in \mathbb{R}^{(B \cdot \Delta T) \times D_{video}^{jca} \times M^{jcv}}$, and $V_{gcn}^{jcp} \in \mathbb{R}^{(B \cdot \Delta T) \times D_{video}^{jcp} \times M^{jcp}}$, respectively, then the attention score for the node can be found using Eqs 9, 10, and 11.

$$Attn_{gcn}^{jcv} = Softmax(FC(V_{gcn}^{jcv})) \tag{9}$$

$$Attn_{gcn}^{jca} = Softmax(FC(V_{gcn}^{jca})) \tag{10}$$

$$Attn_{gcn}^{jcp} = Softmax(FC(V_{gcn}^{jcp})) \tag{11}$$

here, $Attn_{gcn}^{jcv}, Attn_{gcn}^{jca}, Attn_{gcn}^{jcp} \in \mathbb{R}^{(B \cdot \Delta T) \times D_{video}^{jcv} \times 1}$.

The attention-weighted features can be derived using Eqs 12, 13, and 14.

$$V_{gcn-attn}^{jcv} = Attn_{gcn}^{jcv} \odot V_{gcn}^{jcv} \tag{12}$$

$$V_{gcn-attn}^{jca} = Attn_{gcn}^{jca} \odot V_{gcn}^{jca} \tag{13}$$

$$V_{gcn-attn}^{jcp} = Attn_{gcn}^{jcp} \odot V_{gcn}^{jcp} \tag{14}$$

The final output from the GCN can be obtained using Eqs 15, 16, and 17.

$$X_{gcn}^{jcv} = Relu(A @ V_{gcn-attn}^{jcv} @ W + B) \tag{15}$$

$$X_{gcn}^{jca} = Relu(A @ V_{gcn-attn}^{jca} @ W + B) \tag{16}$$

$$X_{gcn}^{jcp} = Relu(A @ V_{gcn-attn}^{jcp} @ W + B) \tag{17}$$

Here, $W \in \mathbb{R}^{M^{jcp} \times M^{jcp'}}$ and $B \in \mathbb{R}^{M^{jcp'}}$ are learnable parameters, $A \in \mathbb{R}^{D_{video}^{jcp} \times D_{video}^{jcp}}$ is the adjacency matrix.

To match the dimension of the output of the attention-based GCN layer with the Bi-LSTM layer, we apply a linear transformation using fully connected (FC) layers on the final outputs of the GCN (Eqs 18, 19, and 20).

$$X_{gcn-t}^{jcv} = Reshape(FC(X_{gcn}^{jcv})) \tag{18}$$

$$X_{gcn-t}^{jca} = Reshape(FC(X_{gcn}^{jca})) \tag{19}$$

$$X_{gcn-t}^{jcp} = Reshape(FC(X_{gcn}^{jcp})) \tag{20}$$

$$X_{student}^{concat,gcn} = \left[ X_{gcn-t}^{jcv}, X_{gcn-t}^{jca}, X_{gcn-t}^{jcp} \right] \tag{21}$$

We also reshape the output from each FC layer and then concatenate features from three modalities to create $X_{student}^{concat,gcn}$ (Eq 21), which in turn helps to match with the dimension of $X_{student}^{concat,bi-lstm}$. This transformation and reshaping help to apply a gating mechanism between Bi-LSTM and the attention-based GCN encoder.

$$X_{student}^{concat,gcn} = [X_{gcn-t}^{jcv}, X_{gcn-t}^{jca}, X_{gcn-t}^{jcp}] \tag{22}$$

**Gated Network:** To effectively combine features from Bi-LSTM and attention-based GCN, we propose a gated network. First, we concatenate the features from both layers. The concatenated features are passed through a fully connected layer with a sigmoid activation function to generate a weight vector for each of the features of $X_{student}^{concat,bi-lstm}$. We subtract the weight vector from one to obtain the weight vector for $X_{student}^{concat,gcn}$. All the steps can be realized by Eqs 23, 24, and 25.

$$X_{student}^{concat,bi-lstm,gcn} = [X_{student}^{concat,bi-lstm}, X_{student}^{concat,gcn}] \tag{23}$$

$$W_{gate} = Sigmoid(FC(X_{student}^{concat,bi-lstm,gcn})) \tag{24}$$

$$X_{student}^{concat,bi-lstm,gcn} = W_{gate} * X_{student}^{concat,bi-lstm} + (1 - W_{gate}) * X_{student}^{concat,gcn} \tag{25}$$

**Multi-Fusion Module (MFM):** We create MFM by slightly modifying our previously proposed multi-modal feature fusion module (MFFM) [27] to better suit the problem settings in this study. We provide the details of MFM implementation in the S1 File. After applying Eqs S1 to S11 to the $X_{student}^{concat,bi-lstm,gcn}$, we obtain $X_{student}^{concat,mfm}$ from Eq 26.

$$X_{student}^{concat,mfm} = MFM(X_{student}^{concat,bi-lstm,gcn}) \tag{26}$$

**Fully connected layers:** Fused features from the MFM are then passed to fully connected layers to produce the output of KFM, KAM, and GRFs (Eq 27).

$$K_{estimation}^{student} = FC(X_{student}^{concat,mfm}) \tag{27}$$

**Teacher model (IMU+Joint Position+MFM+BiLSTM).** The teacher model comprises three Bi-LSTM encoders that are assigned to encode the acceleration, angular velocity data from the IMU, and the 2-D joint center position data from videos. Additionally, it includes the MFM that is similar to the student model, as well as fully connected layers that are used to map the fused features to the KAM, KFM, and 3D-GRFs. Each component of the teacher model is described in the following sections.

**Bi-LSTM:** The stacked Bi-LSTM layer employed in the student model is also utilized in the teacher model. If the batch normalized features from the IMU's accelerometer, gyroscope, and 2-D joint center position data from a video are $I^{acc} \in \mathbb{R}^{B \times \Delta T \times (D_{imu}^{acc} \cdot M^{acc})}, I^{gyr} \in \mathbb{R}^{B \times \Delta T \times (D_{imu}^{gyr} \cdot M^{gyr})}$, and $V^{jcp} \in \mathbb{R}^{B \times \Delta T \times (D_{video}^{jcp} \cdot M^{jcp})}$, respectively, then output from the Bi-LSTM encoder is

$$X_{bi-lstm}^{acc} = Bi - LSTM(I^{acc}) \tag{28}$$

$$X_{bi-lstm}^{gyr} = Bi - LSTM(I^{gyr}) \tag{29}$$

$$X_{bi-lstm}^{jcp} = Bi - LSTM(V^{jcp}) \tag{30}$$

Then, we concatenate features from three modalities to fuse all the multi-modal features together (Eq 31).

$$X_{teacher}^{concat} = \left[ X_{bi-lstm}^{acc}, X_{bi-lstm}^{gyr}, X_{bi-lstm}^{jcp} \right] \tag{31}$$

**MFM:** Concatenated features from teacher encoders $X_{teacher}^{concat}$ are gone through MFM to obtain $X_{teacher}^{concat,mfm}$ using Equation from S1 to S11 from S1 File.

**Fully Connected Layers:** Fused features $X_{teacher}^{concat,mfm}$ from the MFM of the teacher module are passed to a fully connected linear layer to produce the output of KFM, KAM, and GRFs (Eq 32).

$$K_{estimation}^{teacher} = FC(X_{teacher}^{concat,mfm}) \tag{32}$$

## Methods for comparison

**Model ablation (student).** In this subsection, we will discuss all the intermediate models used to compare with the final proposed student model.

• **Joint Position+BiLSTM:** In Joint Position+BiLSTM, we utilize 2-D joint center position data as input and Bi-LSTM as the encoder for kinetics estimation.

• **Augmented Joint Kinematics+Early Fusion+BiLSTM:** The 2-D joint center velocity and acceleration are obtained from the 2-D joint center position data. These features are then combined via early fusion (i.e., concatenation at the input level across features) and passed to the Bi-LSTM encoder.

• **Augmented Joint Kinematics+Concat+BiLSTM:** To enhance the effectiveness of feature extraction, it is beneficial to consider the 2-D joint center position, velocity, and acceleration as different modalities. For this, we employ a Bi-LSTM encoder for each modality individually, then concatenate the features and link them to the output layer to construct Augmented Joint Kinematics+Concat+BiLSTM.

• **Augmented Joint Kinematics+Concat+GCN:** We replace the Bi-LSTM encoder from Augmented Joint Kinematics+Concat+BiLSTM with the GCN to build Augmented Joint Kinematics+Concat+GCN .

• **Augmented Joint Kinematics+Concat+Attn-GCN:** We apply the attention mechanism to the GCN of Augmented Joint Kinematics+Concat+GCN to build Augmented Joint Kinematics+Concat+Attn-GCN.

• **Augmented Joint Kinematics+Concat+BiLSTM+Attn-GCN:** We utilize both Bi-LSTM and attention-based GCN as the encoders in Augmented Joint Kinematics+Concat+BiLSTM+Attn-GCN. Encoded features from both encoders are added together and connected to the output layer to create Augmented Joint Kinematics+Concat+BiLSTM+Attn-GCN.

• **Augmented Joint Kinematics+Concat+Gate+BiLSTM+Attn-GCN:** The gated network is used to adaptively fuse the features from the Bi-LSTM and attention-based GCN encoders, allowing the model to dynamically balance temporal and spatial information from both encoders.

• **Augmented Joint Kinematics+MFM+Gate+BiLSTM+Attn-GCN:** MFM is applied to the concatenated multi-modal features of Augmented Joint Kinematics+Concat+Gate+BiLSTM+Attn-GCN to create Augmented Joint Kinematics+MFM+Gate+BiLSTM+Attn-GCN.

**Model ablation (teacher).** All the intermediate models compared with the final proposed teacher model are discussed in this subsection.

• **IMU+BiLSTM:** We use IMU data as the input to the Bi-LSTM encoder to create IMU+BiLSTM.

• **IMU+Joint Position+Early Fusion+BiLSTM:** We concatenate IMU and 2-D joint center data at the input level (early fusion) and pass the combined features to the Bi-LSTM encoder to build IMU+Joint Position+Early Fusion+BiLSTM.

• **IMU+Joint Position+Concat+BiLSTM:** We treat IMU's accelerations, angular velocities, and joint center position data as different modalities and encode each modality separately with Bi-LSTM encoders. Encoded features from each modality are concatenated and connected to the output layer to create IMU+Joint Position+Concat+BiLSTM.

**Knowledge distillation (Vanilla and layer loss).** We compare our multi-modal knowledge transfer technique with vanilla knowledge distillation (KD) [32] coupled with additional layer loss. We utilize the loss function of Eq 33 to train the student model. The loss function consists of two components ($L_{student}$ and $L_{KD}$). $L_{student}$ is minimizing the RMSE between prediction and ground truth that is measured from the motion capture system, force plates, and musculoskeletal model, where the first component of $L_{KD}$ is minimizing the prediction error between teacher and student model and the second component is minimizing the RMSE between linearly projected features of the encoders (Eq 35).

$$Loss = L_{student} + \alpha \times L_{KD} \tag{33}$$

$$L_{student} = \left[ \frac{1}{N} \sum_{i=1}^{N} (K_{ground\ truth}^{student,i} - K_{estimation}^{student,i})^2 \right]^{\frac{1}{2}} \tag{34}$$

$$L_{KD} = \left[ \frac{1}{N} \sum_{i=1}^{N} (K_{estimation}^{student,i} - K_{estimation}^{teacher,i})^2 \right]^{\frac{1}{2}} + \left[ \frac{1}{N} \sum_{i=1}^{N} (X_{student}^{concat,lin,i} - X_{teacher}^{concat,lin,i}]^2 \right)^{\frac{1}{2}} \tag{35}$$

Here, $N$ is the total number of samples in the training data.

**State-of-the-art models.** In this subsection, we will provide the details of the models that were previously used to estimate kinetics. All models, including the baselines and our proposed method, were evaluated on the same dataset, preprocessing steps, and leave-one-subject-out cross-validation protocol to ensure fairness of comparison. We utilize 2-D joint center position data from two smartphone cameras for all models. To enable comparison between the multi-modal fusion methods LMFN and TFN in Tian et al. [23] and our MFM, we incorporate joint center velocity and acceleration as additional inputs, even though deriving these modalities is one of our contributions and distinct from Tian et al.'s approach. We modify those models slightly to fit our current dataset, as directly using those models may not result in optimal performance, which is necessary for a valid comparison with our proposed method. To facilitate transparency and reproducibility, we have made the full implementation, including SOTA models and our proposed framework, publicly available in our GitHub repository (*https://github.com/Md-Sanzid-Bin-Hossain/Smartphone-Video-based-KAM-KFM-3D-GRFs*).

**(i) FFN [33,34]:** We batch normalized the input data to perform operation similar to standard scaling [31]. The batch-normalized data is input into three fully connected layers, each followed by a ReLU activation. A dropout layer is used after each of the ReLU activations to avoid overfitting. Features from the final dropout layer are flattened and connected to the last fully connected layer for kinetics prediction.

**(ii) Bi-LSTM [34,35]:** We use two bidirectional LSTM layers, with a dropout layer following each one. Features from the last dropout layers are flattened and connected to a fully connected layer for predicting the kinetics.

**(iii) 2D Conv. Network [22]:** The 2D Conv. network primarily consists of convolutional 2D layers. A batch normalization layer and a max-pooling layer are applied after the convolutional layers. A 2D convolutional layer, a batch normalization layer, and a 2D max-pooling layer build a unit for the 2D Conv. network. Four such units are concatenated together, followed by two fully connected layers with ReLU activation. A dropout layer is added after each fully connected layer. Features from the last fully connected layer are flattened and connected to the output layer for prediction.

**(iv) Kinetics-FM-DLR-Net [21]:** Kinetics-FM-DLR-Net was originally implemented in Keras [36] with the Tensorflow [37] backend. As our proposed model is implemented in Pytorch [38] and performance can vary based on the implementation platform [39], we re-implement Kinetics-FM-DLR-Net in Pytorch [38] environment.

**(v) DL-Kinetics-FM-Net [17]:** We also re-implement DL-Kinetics-FM-Net in Pytorch [38], which was originally implemented in Keras [36] with Tensorflow [37] backend.

**(vi) LMFN [23]:** We slightly update the LMFN architecture from [23] to suit the input features of this study. Specifically, a Bi-LSTM layer encodes each modality individually, after which the features are concatenated and passed through a low-rank multi-modal fusion module [40]. The fused features are then processed by a fully connected layer with ReLU activation and later connected to the prediction layer.

**(vii) TFN [23]:** We adopt a similar architecture to TFN used in [23] and slightly modify the encoded features of Bi-LSTM by downsampling to tackle the problem of GPU memory constraints resulting from the tensor multiplication of three encoder features.

## Implementation details

All models included in this study are trained using the PyTorch framework, employing a TITAN Xp GPU (NVIDIA, CA). The Adam optimizer [41] is used in this study, with a learning rate of 0.001. RMSE is utilized as the loss function for all the models. The models were run for a total of 40 epochs, implementing early stopping with a patience epoch of 10. The weight of the best model is saved by selecting the smallest loss value from the validation data. All the models were implemented with a batch size of 64.

## Evaluation procedures

To assess the effectiveness of our models, we use a leave-one-subject-out cross-validation method. This method involves excluding the test subject from the training data. This exclusion is critical for proper validation because including the test subject's data in the training set might lead to improved model performance, as it allows the model to comprehend the specific relationship between IMU data and kinetics for that particular subject. To assess performance, we utilize two metrics: normalized root mean square error (NRMSE) and Pearson correlation coefficient (PCC). NRMSE is expressed as a percentage by normalizing the RMSE to the range (the difference between the maximum and minimum values) of the respective kinetics component, i.e., the experimentally measured GRFs and directly calculated joint moments. NRMSE and PCC values are reported as the mean values averaged across all subjects. We performed Repeated-Measures Analysis of Variance (ANOVA) followed by pairwise post hoc comparisons with Bonferroni correction for NRMSE and PCC separately, using IBM SPSS Statistics (Version 29, IBM, Armonk, NY) to assess whether our proposed method offers a significant improvement in kinetics estimation over other methods. The threshold of $p < 0.05$ was used as the primary criterion for statistical significance, while $p < 0.01$ was additionally reported to indicate results with stronger statistical evidence.

## Results and discussion

### Model comparison (student)

In Table 1, we present the results acquired from different techniques using smartphone-based kinetics estimation. While treating 2-D joint center data as unimodal input data, Joint Position+BiLSTM estimated kinetics with a mean NRMSE of 5.71 and PCC of 0.929. By synthesizing 2-D joint center velocity and acceleration and combining it with the 2-D joint center

**Table 1. Comparison of mean and standard deviation of NRMSE (%) and PCC for smartphone-based estimation methods.** This table presents results for different intermediate models and our final proposed model with knowledge transfer **(KT)**.

| Models | NRMSE(%) | PCC |
|---|---|---|
| Joint Position+BiLSTM | 5.71 ± 0.54** | 0.929 ± 0.017** |
| Augmented Joint Kinematics+Early Fusion+BiLSTM | 5.46 ± 0.56** | 0.936 ± 0.015** |
| Augmented Joint Kinematics+Concat+BiLSTM | 5.09 ± 0.53** | 0.942 ± 0.014** |
| Augmented Joint Kinematics+Concat+GCN | 7.63 ± 0.72** | 0.876 ± 0.030** |
| Augmented Joint Kinematics+Concat+Attn-GCN | 7.45 ± 0.69** | 0.881 ± 0.028** |
| Augmented Joint Kinematics+Concat+BiLSTM+Attn-GCN | 5.11 ± 0.56** | 0.943 ± 0.014** |
| Augmented Joint Kinematics+Concat+Gate+BiLSTM+Attn-GCN | 4.96 ± 0.54** | 0.945 ± 0.014** |
| Augmented Joint Kinematics+MFM+Gate+BiLSTM+Attn-GCN | 4.85 ± 0.58** | 0.946 ± 0.015** |
| **Augmented Joint Kinematics+MFM+Gate+BiLSTM+Attn-GCN+KT** | **4.68 ± 0.53** | **0.951 ± 0.013** |

The bold numbers indicate the highest performance in NRMSE and PCC. ** and * indicate a significant difference in NRMSE and PCC between **Augmented Joint Kinematics+MFM+Gate+BiLSTM+Attn-GCN+KT** and the other models, for $p < 0.01$ and $p < 0.05$, respectively.

position data provide a performance improvement from 5.71 to 5.46 for NRMSE and 0.929 to 0.936 for PCC using only a single encoder. However, when we treat 2-D joint center position, velocity, and acceleration as separate modalities and use individual encoders to encode features separately, we achieve substantial improvement with an NRMSE of 5.09 and PCC of 0.942. We use Bi-LSTM as the encoder for all these models.

The data representing human skeletal joint centers exhibits a graph-like structure. To utilize the graph relation, we employ a graph convolutional network in Augmented Joint Kinematics+Concat+GCN, which results in an NRMSE of 7.63. As we are only predicting kinetic parameters for the right leg, it makes sense to utilize an attention mechanism, which will put more weight on the nodes (joint centers) that are closely related to the right leg. After applying the attention mechanism, NRMSE decreases from 7.63 to 7.45, and PCC increases from 0.876 to 0.881. Our performance is lower compared to the Bi-LSTM-based encoders. This could be attributed to the fact that the graph convolutional network focuses solely on the spatial relationships between different nodes, in this case, joint centers, without accounting for the temporal relationships inherent in the data. Incorporating temporal dynamics is crucial for accurately capturing the sequential nature of the motion data, which Bi-LSTM-based encoders effectively address. To utilize the strength of both encoders, we concatenate the features of both encoders to create Augmented Joint Kinematics+Concat+BiLSTM+Attn-GCN, which provides an NRMSE of 5.11. Combining the two encoders does not improve the results, as using Bi-LSTM encoders in Augmented Joint Kinematics+Concat+BiLSTM already provides an NRMSE of 5.09. To effectively concatenate the features from both encoders, we employ a gating mechanism, which helps to lower the NRMSE to 4.96 and increase the PCC to 0.945.

Later, we apply the MFM to the concatenated features of student encoders, which reduces the NRMSE from 4.96 to 4.85. S1 Table of S1 File compares the mean and standard deviation of all kinetic components for different fusion modules of MFM applied to the student model. Our proposed approach, which integrates LWF, WFF, MHAF, and TMF, shows the highest performance, achieving an NRMSE of 4.85 and a PCC of 0.946. The combination of multiple modules, especially in the LWF+WFF+MHAF+TMF configuration, results in the best performance compared to other combinations. This improvement can be attributed to the complementary strengths of each module, leading to a more comprehensive understanding and estimation of the kinetic components. While the final fusion module (LWF+WFF+MHAF+TMF) does not yield statistically significant improvements in kinetics estimation for the student

model, except for the TMF module, it still enhances performance in terms of both NRMSE and PCC metrics. The lack of statistical significance could be attributed to the inherent nature of the multi-modal joint center position, velocity, and acceleration data. Since the fusion module is designed to handle more diverse modalities, it may not exhibit a substantial performance boost for the student model. This is likely because joint center position, velocity, and acceleration represent a relatively homogeneous set of modalities rather than distinctly varied ones.

Finally, we apply the proposed multi-modal knowledge transfer technique with the help of the teacher model (IMU+Joint Position+MFM+BiLSTM), which reduces the NRMSE from 4.85 to 4.68 and increases the PCC from 0.946 to 0.951. We conduct statistical analyses to assess whether our final proposed model significantly outperforms other intermediate models. The results show a statistically significant improvement in both NRMSE and PCC metrics, with a p-value less than 0.01.

While a performance gap remains between the teacher (NRMSE 3.63) and the proposed model (NRMSE 4.68), this reflects a trade-off between prediction accuracy and practical feasibility, as the proposed model requires only smartphone video input without additional IMU sensors. Despite outperforming existing methods, it is challenging to directly interpret the clinical significance of a specific NRMSE value, as no formal thresholds have been established for joint kinetics to the best of our knowledge. The acceptability of an error margin depends strongly on the clinical context, such as whether the method is applied for early detection, longitudinal monitoring, or individualized clinical decision-making. Nevertheless, our model achieved an NRMSE of 4.68 alongside a correlation coefficient of 0.957 with ground-truth kinetics, which indicates very strong agreement according to established guidelines for correlation interpretation [42,43]. These results suggest that the proposed framework provides highly reliable estimates of joint kinetics, though prospective validation studies will be required to establish task-specific thresholds for clinical acceptability.

We compare the multi-modal knowledge transfer with the vanilla knowledge distillation techniques [32]. We present the results with two different loss functions in S2 Table of S1 File. For KD (Vanilla), we only utilize the first portion of Eq 35, while for KD (Vanilla+layer loss), we use both portions of the loss of Eq 35. We also change the $\alpha$ from 0.10 to 0.90 for each case. No distinctive performance improvement is seen in S2 Table. The proposed knowledge transfer significantly outperforms the majority of cases. Additionally, this knowledge transfer approach does not require any parameter tuning, such as $\alpha$, which reduces the training complexity and time for the model optimization.

Table 2 presents the mean and standard deviation of each component separately, estimated by Augmented Joint Kinematics+MFM+Gate+BiLSTM+Attn-GCN+KT. Our proposed model achieves an NRMSE of 5.03 and PCC of 0.928 for KFM, while NRMSE of 5.86 and PCC of 0.908 for KAM. On the other hand, GRF components, specifically vertical and anterior-posterior GRF, demonstrate excellent accuracy with higher PCC and lower NRMSE.

**Table 2. Component-wise Performance of the Augmented Joint Kinematics+MFM+Gate+BiLSTM+Attn-GCN+KT Model.** The table presents the NRMSE and PCC values separately for knee joint moments and GRF components estimated using the student model.

| Component | NRMSE(%) | PCC |
|---|---|---|
| KFM | 5.03 ± 1.43 | 0.928 ± 0.046 |
| KAM | 5.86 ± 1.65 | 0.908 ± 0.026 |
| Mediolateral GRF | 4.41 ± 0.42 | 0.959 ± 0.010 |
| Vertical GRF | 4.40 ± 0.53 | 0.988 ± 0.009 |
| Anterior-Posterior GRF | 3.69 ± 0.76 | 0.971 ± 0.013 |

Overall, these results suggest that Augmented Joint Kinematics+MFM+Gate+BiLSTM+Attn-GCN+KT is effective in estimating different kinetic components, with particularly strong performance in GRF-related measurements.

We demonstrate plots of two gait cycles in Fig 3, together with the NRMSE and PCC values calculated between the ground truth and the model prediction. Utilizing both qualitative and quantitative comparisons helps in understanding the efficacy of our method for estimating kinetics.

### Model comparison (teacher).

We also perform ablation studies to build the teacher model and present the results in Table 3. Initially, we use IMU data in IMU+BiLSTM, which produces an NRMSE of 4.38 and a PCC of 0.961. While integrating the 2-D joint center with the IMU, the NRMSE decreases from 4.38 to 4.09, and the PCC increases from 0.961 to 0.965. We later consider accelerations and angular velocities from IMUs and 2-D joint center position data as separate modalities and extract features using individual encoders to build IMU+Joint Position+Concat+BiLSTM, which further reduces the NRMSE and increases the PCC. By integrating MFM with the concatenated features of IMU+Joint Position+Concat+BiLSTM to create IMU+Joint Position+MFM+BiLSTM, the NRMSE is further reduced from 3.92 to 3.63, demonstrating improved performance. The IMU+Joint Position+MFM+BiLSTM model demonstrates the best performance, achieving the lowest NRMSE of 3.63 and the highest PCC of 0.971. These results are significantly better than those of the other models, as indicated by the bold

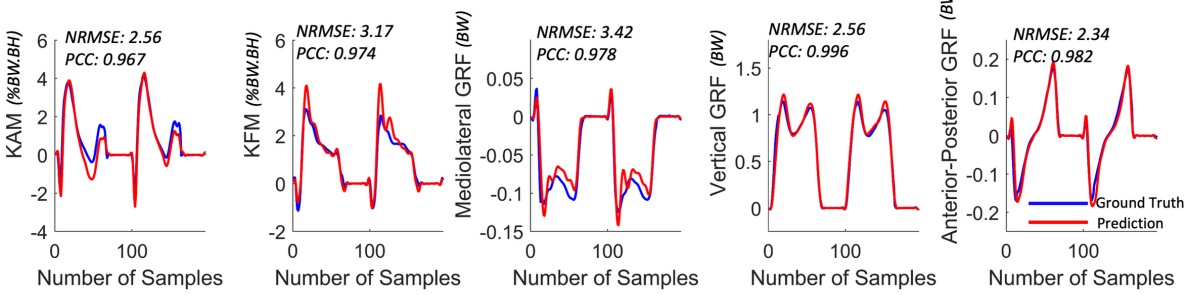

**Fig 3. Sample plots of two gait cycles of the ground truth (blue lines) and prediction (red lines) with NRMSE and PCC to demonstrate quantitative and qualitative comparisons.**

**Table 3. Mean and standard deviation of all the kinetic components for the models utilizing both IMUs and 2-D joint center position data.**

| Models | NRMSE(%) | PCC |
|---|---|---|
| IMU+BiLSTM | 4.38 ± 0.59** | 0.961 ± 0.012** |
| IMU+Joint Position+Early Fusion+BiLSTM | 4.09 ± 0.44** | 0.965 ± 0.010** |
| IMU+Joint Position+Concat+BiLSTM | 3.92 ± 0.42** | 0.968 ± 0.009** |
| **IMU+Joint Position+MFM+BiLSTM** | **3.63 ± 0.47** | **0.971 ± 0.009** |

The bold numbers indicate the highest performance in NRMSE and PCC. ** and * indicate a significant difference in NRMSE and PCC between IMU+Joint Position+MFM+BiLSTM and the other models, for $p < 0.01$ and $p < 0.05$, respectively.

numbers. This underscores the superior accuracy and reliability of the IMU+Joint Position+MFM+BiLSTM model in estimating kinetic components using the combined data from IMUs and 2-D joint center positions.

S3 Table of S1 File presents the mean and standard deviation of all the kinetic components for various fusion modules applied to the teacher model for both NRMSE and PCC metrics. Specifically, our final fusion module (LWF+WFF+MHAF+TMF) achieves the best overall performance with an NRMSE of 3.63 and a PCC of 0.973. This combination outperforms the majority of other fusion module combinations and shows statistically significant improvements. The LWF+WFF+MHAF+TMF module in the teacher model provides a statistically significant improvement over other intermediate fusion strategies. In contrast, the MFM module in the student model does not demonstrate a similarly significant improvement, as discussed earlier. This superior performance is likely due to the teacher model's use of acceleration, angular velocity, and joint center position data, which provide a more dominant and diverse multi-modal nature of the input data compared to the student model's input modalities.

Table 4 presents the mean and standard deviation of each component separately, estimated by the teacher model. The model demonstrates excellent performance, with high accuracy and correlation in estimating joint moments and ground reaction forces. The vertical and anterior-posterior show the lowest errors, indicating strong performance, particularly in GRF components.

## State-of-the-art comparison

Table 5 presents a comprehensive comparison of the performance and number of model parameters between state-of-the-art models and our proposed approach. Our model achieves an NRMSE of 4.68 and a PCC of 0.951 with 4.61 million parameters. These results are superior to those of other models, including the FFN, Bi-LSTM, 2D Conv. Network, Kinetics-FM-DLR-Net, and DL-Kinetics-FM-Net, which reported higher NRMSE and lower PCC values. The significant improvement in both NRMSE and PCC metrics, highlighted in bold, demonstrates the effectiveness of our model. Furthermore, our method has a lower number of parameters compared to the state-of-the-art models, except for the FFN and LMFN. Despite this exception, there exists a considerable performance gap between our proposed model and FFN, indicating that our approach is overall superior in terms of both performance and computational complexity.

While our study builds upon the problem and dataset initially introduced by Tan et al. [23], our methodology and contributions differ significantly in several key areas. Their focus is on the fusion of IMU and smartphone cameras for the estimation of KAM and KFM, whereas our approach promotes a more accessible method by relying solely on smartphone video data. Furthermore, we utilize the augmentation of joint position data to provide a better understanding of Augmented Joint Kinematics and enable us to apply MFM, achieving an

**Table 4. Mean and standard deviation of each component separately from the estimation of the teacher model.**

| Component | NRMSE(%) | PCC |
|---|---|---|
| KFM | 3.88 ± 1.37 | 0.959 ± 0.035 |
| KAM | 4.92 ± 1.68 | 0.949 ± 0.021 |
| Mediolateral GRF | 3.66 ± 0.55 | 0.973 ± 0.006 |
| Vertical GRF | 2.83 ± 0.39 | 0.995 ± 0.005 |
| Anterior-Posterior GRF | 2.87 ± 0.48 | 0.982 ± 0.009 |

**Table 5. Comparison of Augmented Joint Kinematics+MFM+Gate+BiLSTM+Attn-GCN+KT with state-of-the-art methods.** This table compares our model's performance with various existing deep learning approaches, demonstrating its superior accuracy and efficiency.

| Approach | NRMSE(%) | PCC | # Params(M) |
|---|---|---|---|
| FFN [33], [34] | 6.05 ± 0.53** | 0.920 ± 0.015** | 1.79 |
| Bi-LSTM [34], [35] | 5.56 ± 0.55** | 0.931 ± 0.015** | 5.14 |
| 2D Conv. Network [22] | 5.38 ± 0.49** | 0.936 ± 0.013** | 7.06 |
| Kinetics-FM-DLR-Net [21] | 5.18 ± 0.46** | 0.940 ± 0.014** | 77.67 |
| DL-Kinetics-FM-Net [17] | 5.02 ± 0.50** | 0.944 ± 0.014** | 78.18 |
| LMFN [23] | 5.43 ± 0.58** | 0.936 ± 0.018** | 0.68 |
| TFN [23] | 5.15 ± 0.56** | 0.941 ± 0.015** | 5.31 |
| **Ours** | **4.68 ± 0.53** | **0.951 ± 0.013** | 4.61 |

The bold numbers indicate the highest performance in NRMSE and PCC. ** and * indicate a significant difference in NRMSE and PCC between Augmented Joint Kinematics+MFM+Gate+BiLSTM+Attn-GCN+KT and the other SOTA models, for $p < 0.01$ and $p < 0.05$, respectively.

overall NRMSE of 4.85 for all the kinetic components (Table 1), which outperforms both the LMFN and TFN fusion modules , with NRMSEs of 5.43 and 5.15, respectively (Table 5). Additionally, we introduce temporal-spatial modeling of these augmented Augmented Joint Kinematics to further enhance input representation and prediction accuracy. Finally, using knowledge transfer reduces the NRMSE to 4.68 (Table 5), demonstrating even greater improvement.

While using eight IMUs and two smartphones, Tan et al. [23] achieve rRMSE and correlation coefficients of 7.8 and 0.94 for KAM and 6.5 and 0.96 for KFM estimation, while we achieve NRMSE and correlation coefficients of 4.92 and 0.949 for KAM and 3.88 and 0.959 for KFM. For smartphone-only estimation, while Tan et al. [23] provide plots (Cameras Alone) rather than direct numerical values, their results indicate less accurate estimations compared to our smartphone-only approach, which achieves KFM (5.03, 0.928) and KAM (5.86, 0.908).

Although these quantitative results demonstrate the superiority of our approach, a direct comparison with the numerical results reported in Tan et al. [23] should be interpreted with caution due to the difference in experimental setups. For example, our study employs leave-one-subject-out cross-validation to evaluate generalization across individuals, while their study uses five-fold cross-validation, which may introduce variability, complicating direct comparison. Furthermore, while their work focuses exclusively on estimating KAM and KFM, we expand the scope by also estimating 3D GRFs, enabling a more comprehensive assessment of gait kinetics. Despite these differences, we provide a comprehensive comparison to highlight the superiority of our methods.

Our dataset was collected on a treadmill, where continuous gait cycles were captured under controlled conditions. The reported NRMSE of 4.68 reflects performance across all gait cycles included in our analysis, not just a single cycle, suggesting that comparable accuracy can be expected across repeated cycles as long as they are reliably recorded by smartphones. In real-world clinical environments, overground walking is more common, where factors such as walkway length, camera placement, and patient mobility may influence how many complete gait cycles can be recorded. Provided that full cycles are accurately processed by Open-Pose, we anticipate similar performance within the reported error boundary. Future validation under overground and clinical conditions will be important to confirm robustness across multiple gait cycles and practical deployment scenarios.

As OpenPose-based estimation can introduce uncertainty and error into the input data, it is important to consider how this may influence the interpretation of our model predictions.

Prior studies have shown that OpenPose tracking deviates from gold-standard optical motion capture [44], while other works using optical motion capture–derived kinematics to predict kinetics still report non-negligible estimation errors [45]. Together, these findings suggest that the errors observed in our framework likely reflect a combination of (i) uncertainty in smartphone-based input tracking and (ii) inherent model-related limitations.

**Translational impact statement:** This method has a high potential to be translated into the patient monitoring system. Using IMUs or other wearable sensors can be challenging for patients, as it requires them to buy extra devices and continuously wear and maintain them throughout their daily activities. As a result, smartphone-based analysis could provide more convenience for the patients, enabling prompt evaluation of Augmented Joint Kinematics. Although our proposed system is validated with unimpaired individuals, it demonstrates strong potential for application in individuals with knee osteoarthritis, supported by prior machine learning-based studies on accurate knee moment estimation from wearable sensors [46] or with the body key points of marker trajectories [47].

**Limitations:** While this study provides promising results, some methodological limitations must be acknowledged: (1) the dataset used in this study consisted of a relatively homogeneous group of young, healthy males with limited variation in age, height, and weight. This lack of demographic and anthropometric diversity may restrict the generalizability of the deep learning model to broader populations, such as females, older adults, or patients with gait impairments. While utilizing leave-one-subject-out cross-validation mitigates some of the generalization issue for unseen subjects, still a large cohort of subject training is needed for clinical deployment; (2) the outcome from this method largely depends on the accurate extraction of 2-D joint center position data acquired from cameras, which can be sensitive to occlusions, camera placement, and lighting; (3) joint center velocities and accelerations were generated using the finite differencing method without applying any low-pass filtering or smoothing, which may introduce high-frequency noise into the input. Although recurrent models such as LSTMs or GRUs can leverage temporal context and are relatively robust to small amounts of noise [48,49], excessive noise can still degrade model performance; (4) although the proposed architecture outperforms state-of-the-art models in accuracy and efficiency, it may still be too computationally intensive for real-time use on mobile devices without further optimization.

**Future work:** Future studies should address these limitations by incorporating a diverse population comprising different age groups, sexes, and clinical populations to enhance generalizability. To mitigate the effect of high-frequency noise introduced by the finite differencing method, future work could incorporate signal processing techniques such as low-pass filtering or smoothing, or explore noise-robust training strategies. Another key step is to evaluate model robustness under more realistic smartphone recording conditions, including occlusion, varying lighting, and camera quality. To facilitate real-time deployment in clinical and home settings with mobile devices, the architecture can be compressed with advanced deep learning methods such as pruning, quantization, or reducing the number of units in the selected layers. Such strategies are essential to ensure that the framework can be realistically adopted in clinics, where computing resources and time are limited. Moreover, longitudinal studies are needed to assess the ability of the framework to track changes in KAM, KFM, and GRFs over time. This would allow clinicians to monitor patient progress and treatment response over weeks or months, making the framework more useful in rehabilitation and clinical contexts. Another important direction is to investigate which joint centers contribute most to estimating knee moments. For example, KAM is likely influenced most by the relative positions of the hip, knee, and ankle in the frontal plane, whereas KFM may depend more on the positions of the knee and ankle in the sagittal plane. Studying which joint centers carry the most weight

in the prediction would make the model easier for clinicians to interpret and apply. Additionally, the performance disparity between the teacher and student models could potentially be minimized by incorporating additional loss functions between the layers, further enhancing model accuracy. While this study specifically focuses on the knee joint due to its clinical importance, these approaches could also be extended to other joints, such as the hip and ankle, offering a broader understanding of human movement. Finally, validating the method with osteoarthritis patients for clinical reliability and exploring 3D reconstruction techniques from the multi-view cameras for a better understanding of Augmented Joint Kinematics are crucial steps for improved outcomes.

## Conclusion

This paper presents only smartphone video-based estimation of KFM, KAM, and 3D-GRFs. We propose novel techniques that significantly improve the accuracy of smartphone-based kinetics estimation compared to state-of-the-art methods, aiming to enhance the usability of such systems. Our highly accurate smartphone video-based estimation model could provide an alternative approach to addressing the current issues with motion capture systems and force plates. As smartphones are widely available, this suggested method has the potential to be used in clinics and daily life without requiring any additional setup.

## Supporting information

**S1 File. Additional data.** File containing additional equations and tables.
(PDF)

## Author contributions

**Conceptualization:** Md Sanzid Bin Hossain.

**Data curation:** Md Sanzid Bin Hossain.

**Formal analysis:** Md Sanzid Bin Hossain.

**Funding acquisition:** Hwan Choi, Zhishan Guo.

**Methodology:** Md Sanzid Bin Hossain.

**Validation:** Md Sanzid Bin Hossain.

**Visualization:** Md Sanzid Bin Hossain.

**Writing – original draft:** Md Sanzid Bin Hossain.

**Writing – review & editing:** Md Sanzid Bin Hossain, Hwan Choi, Zhishan Guo, Sunyong Yoo, Min-Keun Song, Hyunjun Shin, Dexter Hadley.

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
