## [Decision Letter · Decision Letter 0]

22 Jul 2025

PONE-D-25-25428Knowledge Transfer-Driven Estimation of Knee Moments and Ground Reaction Forces from Smartphone Videos via Temporal-Spatial Modeling of Augmented Joint DynamicsPLOS ONE

Dear Dr. Hossain,

Thank you for submitting your manuscript to PLOS ONE. After careful consideration, we feel that it has merit but does not fully meet PLOS ONE’s publication criteria as it currently stands. Therefore, we invite you to submit a revised version of the manuscript that addresses the points raised during the review process.

**Please be sure to address all reviewer comments for clarifications and additional details. If at all possible, including at least a small pilot study to diversify the subject pool would strengthen the paper.**

We look forward to receiving your revised manuscript.

Kind regards,

Anne E. Martin

Academic Editor

PLOS ONE

**Journal Requirements:**

1. When submitting your revision, we need you to address these additional requirements. Please ensure that your manuscript meets PLOS ONE's style requirements, including those for file naming. The PLOS ONE style templates can be found at https://journals.plos.org/plosone/s/file?id=wjVg/PLOSOne_formatting_sample_main_body.pdf and https://journals.plos.org/plosone/s/file?id=ba62/PLOSOne_formatting_sample_title_authors_affiliations.pdf 2. Please update your submission to use the PLOS LaTeX template. The template and more information on our requirements for LaTeX submissions can be found at http://journals.plos.org/plosone/s/latex. 3. Please note that PLOS ONE has specific guidelines on code sharing for submissions in which author-generated code underpins the findings in the manuscript. In these cases, we expect all author-generated code to be made available without restrictions upon publication of the work. Please review our guidelines at https://journals.plos.org/plosone/s/materials-and-software-sharing#loc-sharing-code and ensure that your code is shared in a way that follows best practice and facilitates reproducibility and reuse. 4. Thank you for stating in your Funding Statement: This work was supported by the National Science Foundation, United States, under Grant FRR-2246671, 2246672.  Please provide an amended statement that declares *all* the funding or sources of support (whether external or internal to your organization) received during this study, as detailed online in our guide for authors at http://journals.plos.org/plosone/s/submit-now.  Please also include the statement “There was no additional external funding received for this study.” in your updated Funding Statement. Please include your amended Funding Statement within your cover letter. We will change the online submission form on your behalf. 5. Thank you for stating the following in the Acknowledgments Section of your manuscript: This work was supported by the National Science Foundation, United States, under Grant FRR-2246671, 2246672. We note that you have provided funding information that is not currently declared in your Funding Statement. However, funding information should not appear in the Acknowledgments section or other areas of your manuscript. We will only publish funding information present in the Funding Statement section of the online submission form. Please remove any funding-related text from the manuscript and let us know how you would like to update your Funding Statement. Currently, your Funding Statement reads as follows:  This work was supported by the National Science Foundation, United States, under Grant FRR-2246671, 2246672. Please include your amended statements within your cover letter; we will change the online submission form on your behalf. 6. Please note that your Data Availability Statement is currently missing the repository name. If your manuscript is accepted for publication, you will be asked to provide these details on a very short timeline. We therefore suggest that you provide this information now, though we will not hold up the peer review process if you are unable. 7. Your ethics statement should only appear in the Methods section of your manuscript. If your ethics statement is written in any section besides the Methods, please move it to the Methods section and delete it from any other section. Please ensure that your ethics statement is included in your manuscript, as the ethics statement entered into the online submission form will not be published alongside your manuscript. 8. If the reviewer comments include a recommendation to cite specific previously published works, please review and evaluate these publications to determine whether they are relevant and should be cited. There is no requirement to cite these works unless the editor has indicated otherwise. 

Reviewers' comments:

Reviewer's Responses to Questions

**Comments to the Author**

1. Is the manuscript technically sound, and do the data support the conclusions?

Reviewer #1: Yes

Reviewer #2: Yes

2. Has the statistical analysis been performed appropriately and rigorously? 

Reviewer #1: No

Reviewer #2: Yes

3. Have the authors made all data underlying the findings in their manuscript fully available?

Reviewer #1: Yes

Reviewer #2: Yes

4. Is the manuscript presented in an intelligible fashion and written in standard English?

Reviewer #1: Yes

Reviewer #2: Yes

5. Review Comments to the Author

**Reviewer #1: **General Overview

Overall, this study presents a smartphone-based kinetics estimation pipeline that integrates augmented kinematic data, knowledge transfer, and spatio-temporal modeling. The diverse machine learning techniques alongside biomechanical insights has led to improved estimation performance, which the authors have validated through rigorous ablations.

However, I have a major concern regarding the generalizability of the proposed pipeline. Although the study includes data from seventeen subjects, the variation in gender, age, height, and weight is extremely limited. This raises doubts about the robustness of the estimation performance, particularly when tested on subjects with gait characteristics similar to those in the training set.

Due to the narrow subject diversity, applying this technology to patients with irregular or pathological gait patterns may result in poor estimation accuracy and unreliable clinical interpretation.

Additionally, the term “joint dynamics” does not seem appropriate for describing the input to the proposed framework. Dynamics typically refers to properties involving mass or inertia, which are not directly captured by IMUs. I recommend using alternative terms, such as “joint kinematics,” to avoid confusion. (Page 3, Line 60; Page 4, Line 85)

Introduction

While IMUs can be used to predict inertial information such as joint moments, they do not directly provide this information. I recommend using more precise terminology—such as “predict” instead of “provide”—to avoid the confusion.

Method

I recommend that the authors include quantitative results comparing their smartphone-based motion tracking to ground-truth kinematic data. Providing motion tracking metrics (e.g., RMSE of joint angle predictions) would help readers understand the system’s inherent accuracy. This distinction is crucial for understanding whether the observed kinetics estimation errors come from the model itself or limitations in the motion tracking input.

The manuscript states that 2D joint positions were normalized to the subject’s height to mitigate size variability. However, joint kinetics inherently depend on body segment lengths and mass distribution. How did the authors recover the true scale required for joint moment estimation?

It is unclear how the authors combined joint keypoints extracted from two different smartphone videos. Was triangulation or another 3D reconstruction method used? This step should be described in the methodology section for reproducibility.

The authors mention that accelerations and angular velocities were derived from positional data. Since IMUs directly measure these quantities, it would be insightful to compare the derived signals with true IMU measurements. I suggest including a comparison plot of smartphone-derived versus IMU-recorded accelerations and angular velocities. This would help readers evaluate how much signal error propagates through the pipeline.

Deriving acceleration and angular velocity from position data is prone to noise. Did the authors apply any signal processing techniques (e.g., low-pass filtering, smoothing, or delaying) to address this? A brief explanation of the data preprocessing pipeline would improve transparency.

Does the proposed pipeline require the subject’s body weight for joint kinetics estimation? If so, how is this information incorporated into the model? If not, can authors comment on how the absence of body weight information might impact the accuracy of the estimated forces and moments.

Regarding comparisons with state-of-the-art (SOTA) models: did the authors evaluate all models using the exact same dataset and evaluation protocol? Clarifying this in the manuscript will enhance the fairness of the comparative results.

Figure 2: For improved readability, I suggest labeling components like attention score, attention-weighted features, and final outputs directly along the arrows with their corresponding symbols in the diagram.

Figure 3: Please include the units for moment and force values.

Discussion

The authors mentioned that the smartphone-based tracking has inherent limitations due to camera placement. From the utility perspective when used in real clinical settings, how many gait cycles can the proposed framework capture within the reported error boundary (NRMSE of 4.68)?

Also, would it be enough to get NRMSE of 4.68 for most of the clinical applications? I would be more helpful to include such information in the discussion part with some references.

**Reviewer #2:** This is a well-written and technically robust manuscript that presents an innovative method to estimate knee joint moments (KAM, KFM) and 3D ground reaction forces (GRFs) using only smartphone video data. The authors introduce a multi-modal knowledge transfer approach from a teacher model (with IMU and video data) to a student model (video-only) for improving estimation accuracy. The proposed method significantly enhances accessibility and usability for gait analysis and rehabilitation purposes, potentially translating to real-world clinical settings.

1. The dataset includes only young healthy male subjects. The lack of diverse populations (e.g., females, older adults, patients) limits generalizability.

2. As acknowledged, 2D joint estimation quality from OpenPose can degrade in real-world conditions (e.g., occlusions, variable lighting), which can impact model performance.

3. While the model outperforms others, the architecture may still be computationally intensive for real-time use on mobile devices without further optimization.

4. Although the knowledge transfer reduces the performance gap, it’s still notable (NRMSE of 4.68 vs. 3.63), and more discussion on real-world acceptability of this gap would be helpful.

5. While the translational potential is mentioned, real-world deployment strategies, clinical thresholds for acceptability, or use in longitudinal tracking are not deeply discussed.

6. Future studies should test on older adults, females, and patients with gait abnormalities to assess model generalizability.

7. Perform experiments under various lighting, occlusion, and video quality conditions to evaluate robustness.

8. Explore lightweight versions of the model for edge computing or real-time deployment on smartphones.

9. Consider testing how well the model tracks changes in KAM/KFM/GRFs over time, particularly in clinical recovery settings.

10. Provide insights into which joint features (e.g., hip vs. ankle) most influence kinetic estimation to improve interpretability for clinicians.

11. Grammar: The writing is clear and grammatically sound.

12. Figures: Ensure all figures (especially Figs 1–3) are high-resolution and clearly labeled. They are essential for understanding model structure and performance.

13. Acronyms: Ensure all acronyms are spelled out at first use (e.g., GRF, GCN, Bi-LSTM).

14. Ethical Statement: Appropriately addressed; no issues.

6. PLOS authors have the option to publish the peer review history of their article (what does this mean?). If published, this will include your full peer review and any attached files.

Reviewer #1: No

Reviewer #2: **Yes: **ARASH MOHAMMADZADEH GONABADI

---

## [Author Response · Author response to Decision Letter 1]

12 Sep 2025

Editor Comments: Please be sure to address all reviewer comments for clarifications and additional details. If at all possible, including at least a small pilot study to diversify the subject pool would strengthen the paper.

Response: We thank the editor for the thoughtful suggestion to include a pilot study with a more diverse subject pool and for emphasizing on addressing all reviewer comments. We have responded to all reviewer comments with extensive clarifications and details.

The present work is based on a publicly available dataset that only includes young, healthy male participants. Since we did not control the data collection process, it is not possible to expand this dataset with additional pilot experiments. Moreover, conducting a new pilot study would require IRB approval and subject recruitment, both of which involve substantial administrative and logistical processes that cannot be readily implemented within the scope of the present study. For these reasons, adding a pilot study is not feasible here, though we fully agree and emphasize in the manuscript that future investigations should recruit more diverse populations to enhance generalizability.

Reviewers' comments:

Reviewer's Responses to Questions

Comments to the Author

1. Is the manuscript technically sound, and do the data support the conclusions?

Reviewer #1: Yes

Reviewer #2: Yes

Response: We thank both reviewers for recognizing that the manuscript is technically sound and that the data appropriately support the conclusions.

2. Has the statistical analysis been performed appropriately and rigorously?

Reviewer #1: No

Reviewer #2: Yes

Response:

We thank both reviewers for their feedback regarding the statistical analysis. We carefully verified our procedures and confirm that all analyses were performed appropriately and rigorously. Specifically, we conducted a Repeated-Measures Analysis of Variance (ANOVA), followed by pairwise post hoc comparisons with Bonferroni correction for both NRMSE and PCC values, using IBM SPSS Statistics (Version 29, IBM, Armonk, NY). In the original manuscript, we mistakenly wrote “least significant difference (LSD)” instead of “Bonferroni correction,” which may have caused confusion. This has been corrected in the revised manuscript. We used p < 0.05 as the primary threshold for statistical significance, and additionally reported p < 0.01 to highlight cases with stronger statistical evidence.

3. Have the authors made all data underlying the findings in their manuscript fully available?

Reviewer #1: Yes

Reviewer #2: Yes

Response: We thank both reviewers for confirming that all data underlying our findings have been made fully available in accordance with PLOS ONE’s data policy.

4. Is the manuscript presented in an intelligible fashion and written in standard English?

Reviewer #1: Yes

Reviewer #2: Yes

Response: We thank both reviewers for acknowledging that the manuscript is clearly written, intelligible, and presented in standard English.

5. Review Comments to the Author

Reviewer #1: General Overview

Overall, this study presents a smartphone-based kinetics estimation pipeline that integrates augmented kinematic data, knowledge transfer, and spatio-temporal modeling. The diverse machine learning techniques alongside biomechanical insights has led to improved estimation performance, which the authors have validated through rigorous ablations.

However, I have a major concern regarding the generalizability of the proposed pipeline. Although the study includes data from seventeen subjects, the variation in gender, age, height, and weight is extremely limited. This raises doubts about the robustness of the estimation performance, particularly when tested on subjects with gait characteristics similar to those in the training set.

Due to the narrow subject diversity, applying this technology to patients with irregular or pathological gait patterns may result in poor estimation accuracy and unreliable clinical interpretation.

Response: We fully agree with the mentioned limitation of the study. We have now further revised with additional context to highlight the limitation of the study for the readers. Specifically, we added the following paragraph at the ‘Limitations’ subsection of the “Results and Discussion” section.

“(1) the dataset used in this study consisted of a relatively homogeneous group of young, healthy males with limited variation in age, height, and weight. This lack of demographic and anthropometric diversity may restrict the generalizability of the deep learning model to broader populations, such as females, older adults, or patients with gait impairments. While utilizing leave-one-subject-out cross-validation mitigates some of the generalization issue for unseen subjects, still a large cohort of subject training is needed for clinical deployment”

Additionally, the term “joint dynamics” does not seem appropriate for describing the input to the proposed framework. Dynamics typically refers to properties involving mass or inertia, which are not directly captured by IMUs. I recommend using alternative terms, such as “joint kinematics,” to avoid confusion. (Page 3, Line 60; Page 4, Line 85)

Response: We agree with the reviewer that the term “joint dynamics” could cause confusion, as dynamics typically refers to mass or inertia-related properties. Therefore, we have replaced “joint dynamics” with “augmented joint kinematics.” We use the qualifier “augmented” to clarify that our features are based on joint center positions, velocities, and accelerations, rather than joint angles as conventionally defined in biomechanics. This terminology has been updated consistently throughout the manuscript, including in the name of the models.

Introduction

While IMUs can be used to predict inertial information such as joint moments, they do not directly provide this information. I recommend using more precise terminology—such as “predict” instead of “provide”—to avoid the confusion.

Response: We sincerely thank the reviewer for pointing this out. We agree that the original sentence seems to be confusing. To address this, we have revised the sentence to improve clarity. Specifically, we removed the previous sentence and added the following sentence in the manuscript: “In contrast, IMUs can measure the segment-level acceleration and angular velocities, which can then be used to predict joint moments.”

Method

I recommend that the authors include quantitative results comparing their smartphone-based motion tracking to ground-truth kinematic data. Providing motion tracking metrics (e.g., RMSE of joint angle predictions) would help readers understand the system’s inherent accuracy. This distinction is crucial for understanding whether the observed kinetics estimation errors come from the model itself or limitations in the motion tracking input.

Response: We thank the reviewer for this valuable suggestion. The dataset used in this study provides optical motion capture marker trajectories and 2D joint keypoints from two smartphone cameras; however, it does not include musculoskeletal modeling outputs such as joint angles from inverse kinematics. Since no ground-truth joint kinematics were provided in the dataset, we cannot directly compute RMSE of joint angles between smartphone-based tracking and marker-based reference data. Nevertheless, prior studies have reported that OpenPose-based tracking introduces measurable errors relative to gold-standard optical motion capture, and other works have shown that even kinetics predicted from MoCap-derived kinematics exhibit non-negligible estimation errors. These findings suggest that the kinetics estimation error observed in our framework reflects a combination of (i) uncertainty in smartphone-based input tracking and (ii) inherent model-related limitations. We have clarified this point in the “Results and Discussion” section of the revised manuscript.

The manuscript states that 2D joint positions were normalized to the subject’s height to mitigate size variability. However, joint kinetics inherently depend on body segment lengths and mass distribution. How did the authors recover the true scale required for joint moment estimation?

Response: We thank the reviewer for pointing this out and agree that further clarification is helpful. In our study, KAM and KFM are provided in dimensionless form, normalized by body weight and body height, and expressed as percentages in the original dataset. We normalized GRFs by subject body weight. These normalization strategies ensure that inter-subject variability due to body size or mass is reduced, enabling the deep learning model to learn subject-independent mappings between 2-D joint center input and kinetics output. Importantly, because the ground truth data are already normalized, the model does not require explicit knowledge of the subject’s absolute body weight or height at inference. If subject-specific values in physical units are needed, they can be recovered post-prediction by rescaling the normalized outputs with the individual’s body weight and height. This approach balances generalizability across subjects while still allowing biomechanically meaningful scaling when subject-specific information is available. We have provided these explanations in the Dataset Description section of the revised manuscript to help readers better understand this point.

It is unclear how the authors combined joint keypoints extracted from two different smartphone videos. Was triangulation or another 3D reconstruction method used? This step should be described in the methodology section for reproducibility.

Response: We thank the reviewer for the feedback. We utilized 2D joint center positions from both cameras and concatenated them into a single feature vector; hence, triangulation or 3D reconstruction was not applied in this work.

The authors mention that accelerations and angular velocities were derived from positional data. Since IMUs directly measure these quantities, it would be insightful to compare the derived signals with true IMU measurements. I suggest including a comparison plot of smartphone-derived versus IMU-recorded accelerations and angular velocities. This would help readers evaluate how much signal error propagates through the pipeline.

Response:

We thank the reviewer for this suggestion. Our input consists of 2D joint center positions, from which we derive linear velocities and accelerations at the joint centers by finite differencing. These are not equivalent to IMU measurements, since IMUs mounted on body segments capture segment-specific linear acceleration (including gravity) and angular velocity in 3D space, which cannot be reconstructed from 2D joint center trajectories without knowledge of segment orientation and sensor pose. For this reason, a direct comparison between derived joint-center dynamics and IMU recordings is not feasible in our dataset.

Deriving acceleration and angular velocity from position data is prone to noise. Did the authors apply any signal processing techniques (e.g., low-pass filtering, smoothing, or delaying) to address this? A brief explanation of the data preprocessing pipeline would improve transparency.

Response: We thank the reviewer for raising this important concern. In our current implementation, we derived velocities and accelerations from joint-center positions using finite differencing without applying additional low-pass filtering. We provide the pre-processing details in the ‘proposed approach’ section (a) Joint center position, velocity, and acceleration).

We acknowledge that this may introduce high-frequency noise; however, recurrent sequence models such as GRUs and LSTMs leverage temporal context, making their predictions more robust to small, noisy variations in the data [1,2].

At the same time, we agree that excessive noise could affect performance, and we have added this point as a limitation in the revised manuscript, noting that future work will investigate noise reduction techniques to further improve robustness.

Specifically, we include the following sentences in our Limitation and Future Work section.

“(3) joint center velocities and accelerations were generated using the finite differencing method without applying any low-pass filtering or smoothing, which may introduce high-frequency noise into the input. Although recurrent models such as LSTMs or GRUs can leverage temporal context and are relatively robust to small amounts of noise [1,2], excessive noise can still degrade model performance.”

“To mitigate the effect of high-frequency noise introduced by the finite differencing method, future work could incorporate signal processing techniques such as low-pass filtering or smoothing or explore noise-robust training strategies.”

1. Yeo K. Short note on the behavior of recurrent neural network for noisy dynamical system. arXiv preprint arXiv:190405158. 2019.

2. Rubinstein B. It’s a super deal–train recurrent network on noisy data and get smooth prediction free. arXiv preprint arXiv:220604215. 2022.

Does the proposed pipeline require the subject’s body weight for joint kinetics estimation? If so, how is this information incorporated into the model? If not, can authors comment on how the absence of body weight information might impact the accuracy of the estimated forces and moments.

Response: We thank the reviewer for raising this concern. Our pipeline does not require explicit body weight during inference, since the ground truth kinetics in the dataset are already provided in normalized form (scaled by body weight and/or height). As a result, the absence of body weight information at inference is not expected to affect model accuracy in the normalized domain. When subject-specific values in physical units are needed, the normalized predictions can be rescaled using the individual’s body weight and height.

We already clarified this in ‘Data Description’ section to help the readers.

Regarding comparisons with state-of-the-art (SOTA) models: did the authors evaluate all models using the exact same dataset and evaluation protocol? Clarifying this in the manuscript will enhance the fairness of the comparative results.

Response: We thank the reviewer for raising this important point. We have clarified in the revised manuscript that all models, including the baselines and our proposed method, were evaluated on the same dataset, preprocessing steps, and leave-one-subject-out cross-validation protocol to ensure fairness of comparison. We also note that full implementations of all baseline models and our proposed framework have been made publicly available in our GitHub repository for transparency and reproducibility.

Figure 2: For improved readability, I suggest labeling components like attention score, attention-weighted features, and final outputs directly along the arrows with their corresponding symbols in the diagram.

Response: We thank the reviewer for this helpful suggestion. We slightly modify Fig. 2 to label attention score, attention weighted features, and final output. Additionally, we al

---

## [Decision Letter · Decision Letter 1]

8 Oct 2025

Knowledge Transfer-Driven Estimation of Knee Moments and Ground Reaction Forces from Smartphone Videos via Temporal-Spatial Modeling of Augmented Joint Kinematics

PONE-D-25-25428R1

Dear Dr. Hossain,

We’re pleased to inform you that your manuscript has been judged scientifically suitable for publication and will be formally accepted for publication once it meets all outstanding technical requirements.

Kind regards,

Anne E. Martin

Academic Editor

PLOS ONE

Additional Editor Comments (optional):

Reviewers' comments:

Reviewer's Responses to Questions

**Comments to the Author**

1. If the authors have adequately addressed your comments raised in a previous round of review and you feel that this manuscript is now acceptable for publication, you may indicate that here to bypass the “Comments to the Author” section, enter your conflict of interest statement in the “Confidential to Editor” section, and submit your "Accept" recommendation.

Reviewer #2: All comments have been addressed

2. Is the manuscript technically sound, and do the data support the conclusions?

Reviewer #2: Yes

3. Has the statistical analysis been performed appropriately and rigorously? 

Reviewer #2: Yes

4. Have the authors made all data underlying the findings in their manuscript fully available?

Reviewer #2: Yes

5. Is the manuscript presented in an intelligible fashion and written in standard English?

Reviewer #2: Yes

6. Review Comments to the Author

Reviewer #2: This is a well-written and technically strong paper with significant potential for advancing accessible gait analysis using consumer-grade devices. Thank you for addressing the comments. it should be good to go for the publication.

7. PLOS authors have the option to publish the peer review history of their article (what does this mean?). If published, this will include your full peer review and any attached files.

Reviewer #2: **Yes: **Arash Mohammadzadeh Gonabadi

---

## [Editor Report · Acceptance letter]

PONE-D-25-25428R1

PLOS ONE

Dear Dr. Hossain,

I'm pleased to inform you that your manuscript has been deemed suitable for publication in PLOS ONE. Congratulations! Your manuscript is now being handed over to our production team.

Kind regards,

on behalf of

Dr. Anne E. Martin

Academic Editor

PLOS ONE